# Multi-Omics Approach Reveals Redox Homeostasis Reprogramming in Early-Stage Clear Cell Renal Cell Carcinoma

**DOI:** 10.3390/antiox12010081

**Published:** 2022-12-29

**Authors:** Wei Zhang, Xinhua Qiao, Ting Xie, Wenbin Cai, Xu Zhang, Chang Chen, Yaoguang Zhang

**Affiliations:** 1Department of Urology, Beijing Hospital, National Center of Gerontology, Institute of Geriatric Medicine, Chinese Academy of Medical Sciences, Beijing 100730, China; 2National Laboratory of Biomacromolecules, CAS Center for Excellence in Biomacromolecules, Institute of Biophysics, Chinese Academy of Sciences, Beijing 100101, China; 3Tianjin Key Laboratory of Metabolic Diseases, Collaborative Innovation Center of Tianjin for Medical Epigenetics, Center for Cardiovascular Diseases, Research Center of Basic Medical Sciences, Department of Physiology and Pathophysiology, Tianjin Medical University, Tianjin 300070, China

**Keywords:** multi-omics, proteomics, redox homeostasis, metabolic reprogramming, clear cell renal cell carcinoma, S-nitrosylation, S-nitrosation

## Abstract

Clear cell renal cell carcinoma (ccRCC) is a malignant tumor originating from proximal tubular epithelial cells, and despite extensive research efforts, its redox homeostasis characteristics and protein S-nitrosylation (or S-nitrosation) (SNO) modification remain largely undefined. This serves as a reminder that the aforementioned features demand a comprehensive inspection. We collected tumor samples and paracancerous normal samples from five patients with early-stage ccRCC (T1N0M0) for proteomic, SNO-proteome, and redox-targeted metabolic analyses. The localization and functional properties of SNO proteins in ccRCC tumors and paracancerous normal tissues were elucidated for the first time. Several highly useful ccRCC-associated SNO proteins were further identified. Metabolic reprogramming, redox homeostasis reprogramming, and tumorigenic alterations are the three major characteristics of early-stage ccRCC. Peroxidative damage caused by rapid proliferation coupled with an increased redox buffering capacity and the antioxidant pool is a major mode of redox homeostasis reprogramming. NADPH and NADP^+^, which were identified from redox species, are both effective biomarkers and promising therapeutic targets. According to our findings, SNO protein signatures and redox homeostasis reprogramming are valuable for understanding the pathogenesis of ccRCC and identifying novel topics that should be seriously considered for the diagnosis and precise therapy of ccRCC.

## 1. Introduction

Renal cell carcinoma is one of the most common urological tumors [1], of which more than 90% of cases are clear cell renal cell carcinoma (ccRCC) originating from the proximal tubular epithelial cells [2]. Metabolic reprogramming is one of the main characteristics of ccRCC [3], and thus ccRCC was once referred to as a metabolic disease [4]. Metabolic reprogramming is a hallmark of malignancy [5], and cancer cells can counter the energy supply shortage and redox homeostasis imbalance caused by their rapid proliferation through metabolic reprogramming [6]. An increasing number of studies have confirmed that malignancies have the propensity to sustain redox homeostasis [7], and recent bioinformatics studies have found that redox-related genes or long non-coding RNAs are associated with the prognosis of ccRCC [8,9]. Although an increasing number of omics studies are being used to profile different cancers, the results obtained from transcriptomics, proteomics, and metabolomics are often divergent [10,11]. Post-translational modifications of proteins are one of the main reasons for this discrepancy. In addition to the more common forms of post-translational modifications such as phosphorylation and ubiquitination, protein S-nitrosylation (or S-nitrosation) (SNO) modifications have gradually been shown to influence a variety of pathophysiological processes [12]. SNO modifications refer to the reversible covalent attachment of cysteine residues of proteins by nitric oxide to form S-nitrosothiols, an alteration that can significantly affect the proper function of proteins [13]. SNO proteins are involved in the pathogenesis of a variety of cancers [14], with potential clinical intervention value. Nevertheless, neither the redox homeostasis signature nor the SNO modification features of ccRCC have been clearly elucidated. Fortunately, the SNO-proteome as well as our previously developed redox environment metabolic evaluation (REME) method provide new opportunities to understand the above characteristics of ccRCC [15].

Using proteomics, the SNO-proteome, and REME technology, the current study examined tumor tissues and adjacent normal tissues from five ccRCC patients. The redox homeostasis signature and the SNO modification features of ccRCC were fully elucidated to provide a foundation for further exploration of ccRCC pathogenesis and precision therapies.

## 2. Materials and Methods

### 2.1. Experimental Design and Tissue Specimens

Tumors and paracancerous normal kidney specimens from five early-stage ccRCC patients (T1N0M0) who underwent surgical treatment at the Department of Urology, Beijing Hospital (Beijing, China) were collected immediately at the time the specimen left the human body. Specimens were rinsed three times with Phosphate Buffered Saline (PBS) and snap-frozen with liquid nitrogen [16]. Kidney specimens from three patients (T1aN0M0) were subjected to quantitative proteomics and SNO-proteome analysis (Figure 1), and kidney specimens from all five patients were subjected to REME. The sex, age, Fuhrman grade, and Tumor Node Metastasis Classification (TNM) stage of the patients are listed in Appendix A. Limited by the volume of kidney specimens, omics testing of each sample was completed with only one technical replicate. All the renal tissue specimens were identified by experienced pathologists. This study was approved by the ethics committee of Beijing Hospital (Beijing, China, No. 2021BJYYEC-282-01) and conducted following the Declaration of Helsinki. Each patient gave written informed consent.

### 2.2. Irreversible Biotinylation Procedure (IBP)

The SNO levels of proteins were detected according to the IBP method as described previously [17]. Briefly, kidney tissues were lysed in HENS buffer (a cell lysis buffer) containing 1% Nonidet P-40, and the supernatant was obtained by centrifugation. S-methylmethanethiosulfonate (MMTS) was added to the supernatant, incubated, and shaken at 50 °C for 30 min. Cold acetone was added to the precipitate and centrifuged (3500× *g*, 10 min). The above steps were repeated 3 times to remove excess MMTS. The precipitates were recovered by incubation in HEN buffer (containing 0.2 mM biotin-maleimide, 2.5% SDS, and 10 mM ascorbate) for 1 h at 37 °C. Excess biotin-maleimide was removed by cold acetone as described previously. The excess biotin-maleimide was likewise removed by cold acetone precipitation. Pellets were resuspended with HEN buffer (containing 200 mM Dithiothreitol (DTT), 2.5% SDS) and incubated at 100 °C for 15 min. Neutralization buffer (100 mM NaCl, 0.1 mM Ethylenediaminetetraacetic acid (EDTA), 10 mM neocuproine, 250 mM Hydroxyethylpiperazine Ethane Sulfonic Acid (HEPES), pH 7.7) and streptavidin agarose (50–100 μL/sample) were added to purify the biotinylated proteins. Then, they were incubated at room temperature for 2 h. The agarose was washed 3 times (centrifugation at 800 g for 1 min) with neutralization buffer (containing 0.05% SDS), and the proteins were finally eluted with HEN buffer (containing 2.5% SDS) at 100 °C for 15 min.

### 2.3. SNO-Proteome

SNO proteins were prepared using the IBP method, and samples were labeled with the iodoTMT kit (Thermo Fisher Scientific, Waltham, MA, USA). As shown in Figure 1, paracancerous renal tissue was labeled 126, 127, and 128, and renal cancer tissue was labeled 129, 130, and 131. Before the addition of 75 μL lysis buffer, proteins were then digested with sequencing-grade modified trypsin. Samples were incubated overnight at 37 °C. The product was dried and suspended with 20 μL TEAB (Triethylammonium bicarbonate) buffer. Mass spectrometry was performed after suspension of the mixed samples with C18 spin-tip desalination. The iodoTMT-labeled peptides were enriched via specific binding to an anti-TMT bead, followed by elution [18].

### 2.4. Liquid Chromatography Triple Quadrupole Mass Spectrometry (LC-MS/MS) Analysis

Q Exactive (Thermo Scientific, Waltham, MA, USA) with an Easy n-LC 1000 HPLC system (Thermo Scientific, Waltham, MA, USA) was used to perform LC-MS/MS experiments. The iodoTMT-labeled peptides were added to a 100 μm id × 2 cm fused silica trap column packed in-house with reversed-phase silica (Reprosil-Pur C18 AQ, 5 μm, Dr. Maisch GmbH, Ammerbuch, Germany) and then separated on a 75 μm id × 20 cm C18 column packed with reversed-phase silica (Reprosil-Pur C18 AQ, 3 μm, Dr. Maisch GmbH, Ammerbuch, Germany). A 78 min duration of the linear gradient was used to elute the peptides attached to the column. Formic acid was present in solvents A and B at a concentration of 0.1% in water and 0.1% in acetonitrile, respectively. At a flow rate of 310 nl/min, the gradient consisted of 4–10% B, 5 min; 10–22% B, 53 min; 22–32% B, 12 min; 32–90% B, 1 min; and 90% B, 7 min. With a mass range of 350–1600 m/z, MS data were collected utilizing the data-dependent acquisition mode at high resolution (70,000; m/z 200). The target value was 3.00 × 10^6^, and the injection time was limited to a maximum of 60 ms. With an isolation width of 2 m/z for fragmentation in the HCD collision cell and a normalized collision energy of 32%, the top 20 precursor ions from each MS full scan were chosen. MS/MS spectra were then obtained at a resolution of 17,500 (m/z 200). The dynamic exclusion time was 40 s, and the goal value was 5.00 × 10^4^, with a maximum injection time of 80 ms. The heated capillary temperature was 320 °C, there was no sheath gas flow, and the spray voltage for the nanoelectrospray ion source setting was 2.0 kV. Two grams of peptides was injected for every analysis, and each sample was evaluated twice [19].

### 2.5. Identification and Measurement of Proteins

The raw data from the Q Exactive were examined using Proteome Discoverer (version 2.2.0.388, Thermo Fisher Scientific, Waltham, MA, USA) and the Sequest HT search engine (Version 1.4.1.14, Thermo Fisher Scientific, Waltham, MA, USA) to determine which proteins were present. The information from cell samples was searched using the Uniprot human protein database (https://www.uniprot.org/ (accessed on 10 May 2022)), which was updated on 10 October 2017. Trypsin was chosen as the enzyme, and two missed cleavages were permitted. Additionally, the mass tolerance of the precursor was set at 10 ppm, and the tolerance for product ions was set at 0.02 Da. Variable modifications included cysteine carbamidomethylation, MMTS, and iodoTMT6plex cysteine and methionine oxidation. The FDR of 1% was established for peptide identification after FDR analysis with Percolator. In Proteome Discovery, protein quantification was also carried out utilizing the ratio of the intensity of the reporter ions from the MS/MS spectra. For protein relative quantitation, only singular and razor-sharp protein peptides were used. The average reported S/N value criterion was set at 10, and the co-isolation barrier was 50% [20].

### 2.6. REME

REME was performed as previously described, and redox environmental metabolites included redox couples (nicotinamide adenine dinucleotide (oxidized form: NAD^+^; reduced form: NADH), nicotinamide adenine dinucleotide phosphate (oxidized form: NADP^+^; reduced form: NADPH), and glutathione (oxidized form: GSSG; reduced form: GSH)); endogenous redox molecules (cysteine (Cys), homocysteine (Hcy), uric acid (UA), and vitamin C (VC)); RNS-related molecules (BH4, L-arginine (Arg), L-citrulline (Cit), and S-nitrosoglutathione (GSNO)); and lipid peroxidation products (4-hydroxy-2-nonenal (4-HNE), 4-hydroxy-2-hexenal (4-HHE), crotonaldehyde (Cro), benzaldehyde (BEN), hexanal, malondialdehyde (MDA), 2,4-nonadienal (2,4.-Non), and 2,4-decadienal (2,4.-Dec)). Briefly, liquid–liquid extraction was used to extract renal tissue samples (50 mg). GSH-13C215N1 (1 μL, 1 mg/mL) and 4-HNE-d3 (1 μL, 1 mg/mL) were added to 400 μL of 75% methanol to make the tissues homogeneous. The homogenate was combined with methyl tert-butyl ether (MTBE; 1 mL), which was added along with 250 μL of water, and the mixture was allowed to stand until it separated. The materials were split into the upper organic phase and the lower aqueous phase by centrifuging them at 12,000× *g* for 10 min at 4 °C. The water-soluble redox molecules in the aqueous phase were immediately transferred to a 0.22 mm centrifuge tube and filtered there (12,000× *g* for 10 min at 4 °C). The top organic phase (fat-soluble lipid peroxidation products) was transferred to a new tube and evaporated to dryness. Next, 75% methanol (500 μL) and 2,4-dinitrophenylhydrazine (500 μL) were added, and the mixture was placed in an oven at 60 C for derivatization for two hours. One milliliter of ethyl acetate was added and thoroughly stirred. The aqueous phase was once more extracted while the organic phase was moved to a fresh tube. The mixed organic phase was dried by evaporation and then dissolved in 100 μL of 60% ACN. Samples were thoroughly mixed and then filtered through a 0.22 μm centrifuge tube before being placed into the mass spectrometer. On an ultra-performance liquid chromatography (UPLC) BEH T3 column (Waters; 1.7 μm, 100 × 2.1 mm i.d.) with a flow rate of 0.3 mL/min, the renal tissue samples were separated. The sample chamber was kept at 4 °C, the column was kept at 25 °C, and the injection volume was set to 10 μL. Water containing 0.25 mM di-n-butylamine acetate (DBAA; solution A) and ACN containing 3 mM DBAA were the polar compound mobile phases (solution B). The gradient was 0–6.5 min, 0–35% B; 6.5–7 min, 35–0% B; and 7–8 min, 100% A. ACN and water with 0.025% acetic acid were the lipid peroxide mobile phases (solution B). The gradient was 0–1 min at 60% B; 1–8 min at 60–100% B; 8–9 min at 100% B; 9–9.5 min at 100–60% B; and 9.5–11 min at 60% B. A hybrid triple quadrupole linear ion trap mass spectrometer (5500 QTRAP, AB SCIEX, Foster City, CA, USA) fitted with a turbo ion spray electrospray ionization source working in negative mode was used to carry out targeted profiling of metabolites. Multiple reaction monitoring (MRM) was used to detect the analytes, and the mass spectrometer was run using Analyst 1.6.1 software (AB SCIEX). CUR = 40 pressure, GS1 = 33 psi, GS2 = 33 psi, IS = 4500 V, CAD = MEDIUM, and TEMP = 500 °C were the ion source settings.

### 2.7. Data Processing and Analysis for Proteomics and SNO-Proteome

Proteome and SNO-proteome data preprocessing was performed using Perseus software (v1.6.15.0, Max Planck Institute of Biochemistry, Martinsried, Germany) to filter out proteins with expression in at least 1 more sample per group and ≥2 unique peptides [21]. Protein or SNO protein expression matrices were normalized based on the median value of each sample and then log2-transformed for further analysis [22]. The minimum value throughout the proteome or glycoproteome data was used to impute missing values. The R package “limma” was used to screen differentially expressed proteins and SNO peptides [23]. Filtering criteria were *p*-value < 0.05 and absolute log2 fold change (log2FC) ≥ 1. In parallel, the Gene Set Enrichment Analysis (GSEA) software (v4.2.3, Broad Institute, Inc., Waltham, MA, USA) was used to search for signature proteins in ccRCC compared with controls [24]. Using the ClueGO (v.2.5.9, Free Software Foundation, Inc. Boston, US) plug-in of the Cytoscape software (v.3.8.2, Free Software Foundation, Inc. Boston, US), analysis of the cellular localization and molecular function of proteomics and the SNO-proteome was carried out based on Gene Ontology (GO) cellular component terms [25]. GO enrichment analysis was performed using the “clusterprofiler” R package [26]. Most of the bioinformatics analysis was conducted using R software (version 3.5.4, https://www.R-project.org) (accessed on 10 May 2022).

The iCysMod database (http://icysmod.omicsbio.info/ (accessed on 10 May 2022)) provided information on the cysteine S-nitrosylation modifications that are currently known [27]. The R package “GSVA” was used to compute protein pathway signatures between the ccRCC and control groups. Background KEGG gene sets (c2.cp.kegg.v7.5.1.symbols, Broad Institute, Inc., Waltham, MA, USA) were downloaded from the Molecular Signatures Database (MSigDB, v7.1, https://www.gsea-msigdb.org/gsea/index.jsp) (accessed on 10 May 2022).

### 2.8. REME Data Processing and Analysis

REME’s LC-MS/MS data processing was performed using Analyst 1.6.1. Multivariate statistical analysis was performed using MetaboAnalyst 3.0 (http://www.metaboanalyst.ca) (accessed on 10 May 2022) and SIMCA software (version 14.1, Sartorius Croatia, Göttingen, Germany). Briefly, raw data were log-transformed (base 10) for further analysis. Supervised discriminant analysis of kidney samples using principal component analysis (PCA) and orthogonal partial least squares discriminant analysis (OPLS-DA) established a relationship between metabolite content and two sets of kidney samples. Promising biomarkers were screened by calculating variable importance for the projection (VIP). Correlations between metabolites as well as between samples were measured by Pearson correlation coefficients. Finally, classical univariate ROC curve analysis was applied to screen the biomarkers used for diagnosis. 

## 3. Results

### 3.1. Proteomic and SNO-Proteome Landscape of ccRCC

We collected five pairs of renal cancer tissues and paracancerous normal tissues according to stringent criteria and performed quantitative proteomic and SNO-proteome analyses of kidney samples from three ccRCC patients staged T1aN0M0 (Figure 1 and Appendix A). In total, 2068 proteins were identified from kidney samples by proteomics (Appendix A). The SNO-proteome identified a total of 636 SNO-cysteine sites localized in 452 proteins, with an average of 444 proteins and 628 SNO-cysteine sites identified per sample (Figure 2A,B, Appendix A). In addition to 325 proteins that were already confirmed to be modified by SNO by the iCysMod database, 127 proteins and their cysteine sites that could be modified by SNO were newly identified in this study (Figure 2C). The number of SNO-cysteine sites varied between proteins, with 352 proteins identified with one SNO-cysteine site, 72 proteins identified with two SNO-cysteine sites, 15 proteins identified with three SNO-cysteine sites, and the remaining 13 proteins identified with four or more SNO-cysteine sites (Figure 2D, Appendix A). As shown in the Venn diagram, 320 proteins were identified in both the proteome and SNO-proteome (Figure 2E, Appendix A). The results of PCA analysis showed that both the proteome and SNO-proteome were able to significantly separate the tumor group from the paracancerous control group (Figure 2F,G). Meanwhile, a sample correlation assessment of the proteome indicated that the correlation coefficients between the ccRCC tumor group samples (0.971, 0.975, and 0.982) were slightly lower than those between the paracancerous control group samples (0.99, 0.991, and 0.992), while the lowest correlation coefficients were found between the ccRCC tumor group and the paracancerous control group (0.846–0.889) (Figure 2H, Appendix A). 

We further performed a comprehensive analysis of the cellular localization and molecular function of all the identified proteins in the kidney tissues. First, SNO proteins differ from protein prototypes in terms of cellular localization. SNO proteins were mainly localized in the redox enzyme complex, accounting for more than 17.65%, followed by extracellular organelles and actin filament bundles, both accounting for 11.76%; in addition, both the ruffle membrane and side of the membrane accounted for 8.82%. Overall, proteins localized to the redox enzyme complex, extracellular organelles, or cell membrane are core targets of SNO modification (Figure 3A and Appendix A, Appendix A). Comparatively, the proportion of redox enzyme complexes in the cellular localization of protein prototypes was not evident (Figure 3B and Appendix A, Appendix A). The results of the functional analysis of SNO proteins showed that SNO proteins in kidney tissues were mainly involved in redox-regulated processes, including oxidoreductase activity, aldehyde dehydrogenase (NAD^+^) activity, aldehyde dehydrogenase (NAD(P)^+^) activity, and electron transfer activity. In addition to these, SNO proteins of kidney tissues were enriched in cell migration functions, including cadherin binding, cell adhesion molecule binding, extracellular matrix structural constituent, and peptidase regulator activity (Figure 4A,B, Appendix A). The results of the functional analysis of the proteome showed similarity to the SNO-proteome, and redox-related molecular functions were similarly enriched, such as oxidoreductase activity, NAD binding, and electron transfer activity. Moreover, invasion/metastasis-related molecular functions, such as cadherin binding, cell adhesion molecule binding, actin binding, actin filament binding, translation factor activity, and translation initiation factor activity, were also the molecular functions with major enrichment in the proteome (Figure 4C,D, Appendix A). The redox functional signatures exhibited by proteomics together with the SNO-proteome suggest a definite value in deeply exploring redox homeostasis changes in ccRCC.

### 3.2. Signature Proteins and SNO Peptides in ccRCC Tissues

To better understand the protein expression profile of ccRCC, we identified signature proteins as well as SNO proteins (SNO peptides) that were differentially expressed between the ccRCC and control groups. A total of 318 differentially expressed proteins and 164 differentially expressed SNO peptides of 126 proteins were found (Appendix A, Figure 5A,B). GO enrichment analysis was further performed on the differentially expressed proteins and SNO proteins to help clarify the corresponding cellular component (CC), molecular function (MF), and biological process (BP) potentially involved in these differential proteins. As shown in Figure 5C, the differential proteins were mainly located in energy-metabolically active and redox-active organelles, including the respiratory chain complex, mitochondrial matrix, respirasome, mitochondrial respirasome, oxidoreductase complex, and inner mitochondrial membrane protein complex. Their main MF, therefore, is to govern the homeostasis of redox and energy metabolism, including electron transfer activity, oxidoreduction-driven active transmembrane transporter activity, NADH dehydrogenase activity, and NADH dehydrogenase (ubiquinone) activity. Additionally, functions for substance transport (primary active membrane transporter activity) were also enriched. Consistent with the preceding findings, the energy metabolism process is the main BP that differential proteins are involved in, including the generation of precursor metabolites and energy, aerobic respiration, cellular respiration, the ATP metabolic process, the electron transport chain, and the aerobic electron transport chain (Appendix A).

As shown in Figure 5D, the results of the GO analysis of the differentially expressed SNO proteins shared similarities with those of the differential proteins and also exhibited their own characteristics. The CC in which differentially expressed SNO proteins reside mainly included the cellular energy metabolism center (mitochondrial matrix) and the center of substance anabolism (endoplasmic reticulum lumen). In addition, organelles that exercise a secretory function (secretory granule lumen, cytoplasmic vesicle lumen, and vesicle lumens) as well as the extracellular matrix (collagen-containing extracellular matrix) were also enriched. The main enriched MF included the activities of redox enzymes (oxidoreductase activities, activities on the CH-CH group of donors), the regulation of substance energy metabolism (electron transfer activity and lyase activity), fatty acid synthesis (acid thiol ligase activity), the construction of the ECM (extracellular matrix structural constituents), and the substance transmembrane transport function (primary active transmembrane transporter activity). Ultimately, all the enriched BPs were focused on the process of material energy metabolism (generation of precursor metabolites and energy, small molecular catabolic process, organic acid catabolic process, carboxylic acid catabolic process, ATP metabolic process, and purine-containing compound metabolic process) (Appendix A).

Based on the above findings, to further define the changing trend of molecular pathways in ccRCC, we performed GSVA analysis on the two groups of samples, and Figure 5E demonstrates the difference between the ccRCC group and the control group. First, the pathways related to substance and energy metabolism were significantly downregulated in the ccRCC group relative to the control group, which included not only the metabolic pathways of fatty acids and various types of amino acids but also the citrate cycle (TCA or tricarboxylic acid cycle) and oxidative phosphorylation (OXPHOS) pathway, as well as the redox-active organelles (peroxisome and lysosome). In addition, the pathways that were significantly downregulated in the ccRCC group included pathways related to normal renal function (proximal tubular bicarbonate reclamation) as well as three neurological-disease-related pathways (Parkinson’s disease, Alzheimer’s disease, and Huntington’s disease). Meanwhile, the significantly upregulated pathways in the ccRCC group mainly included pathways related to the cell cycle of cancer cells (cell cycle), mRNA-translation-related pathways (ribosome), and pathways related to the tumor inflammatory immune microenvironment (Toll-like receptor signaling pathway). Other upregulated pathways not listed in the figure also included pathways in cancer and focal adhesion pathways (Appendix A). The results of the GSEA analysis further confirmed the reliability of the GSVA results (Figure 5F, Appendix A). The upregulated pathways in the ccRCC group mainly involved organelles undergoing the splicing (spliceosome) and translation (ribosome) processes of mRNA, protein catabolic processes (ubiquitin-mediated proteolysis), tumor invasion functions (focal adhesion), and inflammation immune microenvironment remodeling-related pathways (Wnt signaling pathway and chemokine signaling pathway). In line with the desired results, the downregulated pathways in the ccRCC group were largely consistent with the results of GSVA, which mainly focused on the pathways related to substance and energy metabolism.

### 3.3. Molecular Patterns of Independently Differentially Expressed SNO Proteins

To minimize the effect of prototypical expression levels on the degree of SNO proteins, we excluded 64 proteins whose SNO modification levels were consistent with the trend of altered prototype levels. A total of 62 proteins with independent changes in SNO modification levels (containing 81 SNO-cysteine sites) were finally selected for further analysis (Appendix A, Figure 6A). The heatmap in Figure 6B demonstrates the intergroup differences in the SNO-cysteine sites of the above proteins. Differentially expressed SNO proteins were subjected to pathway–protein interaction network module analysis using the ClueGO application in Cytoscape software to clarify the possible impact of SNO modification on ccRCC. Among the enriched network modules, metabolism, OXPHOS, and ECM receptor interactions occupied the core positions (Appendix A, Figure 6C).

### 3.4. REME Profiled the Redox Homeostasis Reprogramming of ccRCC

The enrichment of redox function demonstrated by the proteomic and SNO-proteome analyses and the enrichment of SNO proteins in redox-related organelles suggest a possible remodeling of redox homeostasis in ccRCC that warrants further exploration. Therefore, we detected the 22 most representative redox metabolites in ccRCC tissues and paracancerous normal tissues using our own developed REME method (Appendix A, Appendix A). First, we performed PCA and OPLS-DA analyses of redox metabolic profiles and searched for core redox metabolites in ccRCC using two different clustering methods. The results of the PCA in unsupervised mode showed that the redox components of the ccRCC and control groups had better intergroup differences, and the first two principal components explained 88.1% of the variance in the variables, including PC1 = 72.2% and PC2 = 15.9% (Figure 7A). Furthermore, we performed a supervised OPLS-DA analysis for a more accurate screening of differential redox metabolites. The results showed that the cumulative interpretation rate of the OPLS-DA analysis was sufficient to assess orthogonal components (Appendix A) and that the levels of redox metabolites were significantly different between groups. In addition, the differences in the redox metabolite content of samples within the ccRCC group were significantly greater than those of the control samples (Figure 7B and Appendix A). The results of the sevenfold cross-validation confirmed that the results of OPLS-DA were relatively reliable (R^2^X = 0.823, R^2^Y = 0.861, and Q^2^ = 0.689; Figure 7C). Moreover, after the permutation test (permutation = 999), the regression line intercept of the Q^2^ value of the OPLS-DA model was less than 0.05 (Q^2^ = −0.899, Appendix A). Taking the above results together, we believe that the analytical model adopted in this study has a good fit and prediction ability, and there is no overfitting phenomenon. VIP scores were used to assess the contribution of different redox metabolites to the differences in metabolic profiles, with NADPH, NAD^+^, and NADP^+^ consistently ranking in the top three (Appendix A, Figure 7D and Appendix A). Figure 7E demonstrates the expression pattern of redox metabolites through the form of a heatmap (red for increased abundance and blue for decreased abundance). Performing unsupervised hierarchical clustering on metabolites and samples separately, both the ccRCC group and the control group were able to be accurately distinguished, and most redox metabolites were upregulated in the ccRCC group. Figure 7F (Appendix A) demonstrates the Pearson correlation between samples, and we found that the intraclass correlation was significantly higher for samples from the control group compared with those from the ccRCC group (0.846–0.976 vs. 0.785–0.946), while the correlation between the control group and the ccRCC group was lower (0.622–0.965). Pearson correlation analysis between metabolites showed that most of the lipid peroxidation products (4-HNE, 4-HHE, BEN, hexanal, MDA, 2,4.-Non, and 2,4.-Dec) were poorly correlated, whereas redox couples (NADH/NAD^+^, NADPH/NADP^+^, and GSH/GSSG), endogenous redox molecules (Cys, Hcy, UA, and VC), and RNS-related molecules (BH4, Arg, Cit, GSNO) as well as Cro showed a better correlation (Appendix A, Figure 7G). Finally, we screened out promising biomarkers utilizing ROC curves. Most redox metabolites showed good predictive potency (18/22, AUC > 0.8; 14/22, *p* < 0.05), with NADH/NAD^+^ and NADPH/NADP^+^ (two redox couples) exhibiting the best predictive potential (AUC: 0.92–1; Appendix A, Figure 7H).

## 4. Discussion

In this study, we performed multi-omics on five pairs of ccRCC tumors and adjacent control tissues, including proteomic, SNO-proteome, and redox-targeted metabolic analyses, in an attempt to elucidate the biological characteristics of ccRCC from new perspectives. Our current research is a seminal investigation of the SNO modification pattern of proteins in ccRCC and is also the first to target a representative redox metabolite in ccRCC using the new REME approach. Collectively, the present work fills a gap in the study of protein SNO modifications in ccRCC and adds new research evidence to the shifting landscape of redox homeostasis in ccRCC.

SNO modification of proteins has been widely documented to play an important role in several diseases such as cancer [12,13,14], and our study fills a gap in the study of protein SNO modification in ccRCC. In this study, we found that 636 cysteine sites of 452 proteins were S-nitrosylated in ccRCC tissues and paracancerous normal kidney tissues, which is significantly higher than the proportion in pancreatic cancer tissues [28], implying that SNO modification is a very common type of post-translational modification in kidney tissues. Previous studies have found that SNO-coenzyme A can protect the kidney from acute injury [29]; therefore, we speculated that SNO modification of proteins may be largely involved in the regulatory process of the pathophysiological mechanism of the kidney. Additionally, SNO proteins differ from protein prototypes in their cellular localization. The proportion of SNO proteins localized to the redox enzyme complex exceeds 17%, which suggests the possibility that the more intense the redox response, the more readily proteins are modified by SNO. Further, we found that the pathways enriched in SNO proteins mainly included redox enzyme activity, NAD^+^/NAD(P)^+^ activity, and electron transfer activity. The regulation of redox homeostasis, which is a common intersection of the above three pathways, prompted us to further explore the relevance of SNO modification to redox homeostasis in ccRCC. 

The SNO modification atlas of ccRCC that we describe comprises a number of intriguing targets in addition to providing an overall architecture. For instance, we noticed that the SNO level of succinate dehydrogenase complex subunit A (SDHA) was significantly decreased in the ccRCC group. As an important player in the TCA cycle, SDHA is one of four subunits (including SDHA, SDHB, SDHC, and SDHD) that make up the mitochondrial succinate dehydrogenase complex [30]. Despite the fact that SDHA, a tumor suppressor protein, has been demonstrated to be a contributor to ccRCC, it is still unknown how the SNO modification affects SDHA’s functional status [31]. The only available study has demonstrated that antioxidant cocktail therapy can inhibit the decrease in both prototypical and SNO levels of the SDHA protein in patients with muscle disuse atrophy, indicating a potential protective benefit of an elevated SNO level of SDHA [32]. Therefore, the correlation between the SNO level of SDHA and its functional status in ccRCC has potential research value. Among the numerous proteins that undergo increased levels of nitrosylation modifications in ccRCC, we found two highly valuable candidates: albumin (ALB) and ERO1-like protein alpha (ERO1A). First of all, earlier research has shown that SNO-ALB in the blood might induce vasodilation and prevent platelet aggregation [33]. Whether the upregulated SNO-ALB in ccRCC can promote tumor progression by increasing tumor blood supply is therefore a very promising direction of investigation. At the same time, as described in our previous study, endoplasmic reticulum (ER) dysfunction in senescent cells was associated with decreased oxidase activity caused by increased levels of SNO-ERO1A [34]. Elucidating whether increased SNO-ERO1A contributes to ER dysfunction in normal kidney epithelial cells and thereby causes ccRCC is also an attractive project. Collectively, we demonstrate the ability of the SNO-proteome to serve as a complement to proteomics, providing new evidence and targets to explore the pathogenesis of ccRCC.

Because the processes of transcription, translation, and material metabolism are affected by many factors, the changes in mRNA, proteins, and metabolites are not completely consistent. For example, proteomic data identified significant downregulation of OXPHOS-related proteins in ccRCC tissues, and large-scale transcriptomic studies of ccRCC failed to identify downregulation of OXPHOS-related genes at the mRNA level [10,11]. Therefore, single-omics studies, especially transcriptomics, tend to have limitations in delineating the disease atlas. To provide as comprehensive a picture as possible of the redox homeostasis characteristic of ccRCC, our study revealed the changing trends of enzymes catalyzing the metabolism of substances using proteomic data, on the one hand, and the alterations in major redox metabolites using the method of REME, on the other. The results of the multi-omics analysis revealed three major features of early-stage ccRCC, namely, metabolic reprogramming, redox homeostasis reprogramming, and tumorigenic alterations.

The tumor microenvironment, which is mainly characterized by insufficient nutrients, lack of oxygen, increased metabolic demand, and oxidative stress, hampers rapid tumor proliferation, and for a long time transcriptomic, proteomic, and metabolomic data have suggested that ccRCC can combat the unfavorable tumor microenvironment using metabolic reprogramming [10,11,35,36]. Our study illuminates in detail the metabolic reprogramming exhibited by early-stage ccRCC with alterations in amino acid metabolism, fatty acid metabolism, and energy metabolism as major features. Proteomics found that most of the amino acid and fatty acid metabolic pathways were downregulated in the ccRCC group, which is compatible with the results of previous studies [11,36]. One of the important features of metabolic reprogramming occurring in tumor cells is the Warburg effect, i.e., a decline in TCA and OXPHOS and an increase in glycolysis [37]. The current study confirmed that the energy metabolic processes represented by OXPHOS and the TCA cycle were suppressed in the ccRCC group. The above findings, on the one hand, demonstrate the reliability of the results of the present study [10,11,35,36] and, on the other hand, suggest promising prospects for the treatment of ccRCC through the regulation of material metabolic processes.

In addition to metabolic reprogramming, we found significant redox homeostasis reprogramming in ccRCC. First, a large number of SNO proteins are localized to the redox enzyme complex. Second, proteome and SNO-proteome data showed that the most redox-active organelles such as mitochondria, the ER, and redox enzyme complexes were broadly enriched. Most importantly, our REME results confirmed that the major redox metabolites exhibited distinct profiles in the ccRCC group vs. the control group, and almost all redox metabolites were significantly elevated in the ccRCC group. In general, we speculate that, through redox homeostasis reprogramming, ccRCC cells improve their own redox buffering capacity (e.g., redox couples) and antioxidant material reserves (e.g., endogenous redox molecules) to combat oxidative stress damage, ultimately maintaining their own redox homeostasis at a brand-new, superior level.

First, redox couples (NADH/NAD^+^, NADPH/NADP^+^, and GSH/GSSG) were significantly upregulated in the ccRCC group, and the former two pairs were good biomarkers for ccRCC. As a key molecule in remodeling the redox homeostasis of tumor cells [38], upregulation of NADPH is foreseeable. NADPH can maintain GSH and thioredoxin levels, on the one hand [39,40], and provide a reduction potential for reductive anabolic processes, on the other [41], which provides a material reserve for the rapid proliferation of tumor cells [42]; therefore, it makes sense that tumor cells have higher NADPH levels than normal cells [37,43]. Meanwhile, the mechanisms responsible for NADPH upregulation could be complex. It is currently believed that the pentose phosphate pathway (PPP) and folate-mediated one-carbon metabolism are the main sources of NADPH for cancer cells [44,45,46]. Previous studies have found that the main mechanism by which various cancer cells upregulate NADPH is by increasing glucose entry into the PPP [47,48], and as expected, PPP upregulation is a typical manifestation of metabolic reprogramming in ccRCC [49,50]. Additionally, folate-mediated one-carbon metabolism is also one of the pathways for the generation of NADPH, and the main source of the one-carbon unit is serine [51], which can promote tumor cells’ NADPH production and proliferation ability [52,53]. In vitro experiments have shown that activating the biosynthesis of serine can promote the proliferative capacity of ccRCC cells [54] and that upregulated serine metabolism is associated with poor prognosis in RCC [55]. Similar to NADPH, NAD^+^ also plays an important role in energy metabolism (e.g., the glycolytic process) and redox-regulated processes [56], and its upregulation may likewise contribute to the maintenance of energy supply and redox homeostasis in ccRCC cells. GSH/GSSG is an important redox couple that maintains cellular redox homeostasis, in addition to NADH/NAD^+^ and NADPH/NADP^+^. We found that both GSH and GSSG were upregulated in ccRCC, which is valuable for combating oxidative stress damage and also fits with previous findings [57]. Meanwhile, the upregulation of GSH/GSSG is associated with the grade, stage, and metastasis of tumors [35,57]. In addition, it can reduce the resistance of ccRCC cells to oxidative stress injury by reducing glutamine (the major source of GSH/GSSG) in ccRCC [58], and the clinical application of related medications has been reported to improve the efficacy of ccRCC patients [59].

Second, endogenous redox molecules (Cys, Hcy, VC, and UA) that were upregulated in the ccRCC group are also well defined. Cys is irreplaceable for redox homeostasis maintenance, meaning that cancer cells spend a lot of effort maintaining the Cys pool [60]. Targeting Cys metabolism has a tumor suppressor effect in a variety of cancers [61,62], and metabolomics studies have also confirmed the contribution of increased Cys-related metabolites to ccRCC [35]. Additionally, our study confirms that Hcy is upregulated in ccRCC tissues, a phenomenon consistent with the findings of Qu et al. [36], who identified Hcy-associated homocysteinylation as an important contributor to the malignant phenotype of ccRCC. There are relatively few comparative data on VC in the tumor tissues or blood of ccRCC patients compared with normal individuals, and some studies have pointed out that increasing VC intake can reduce the risk of RCC [63]; however, there are also studies that have found a benefit of intravenous application of high-dose VC in RCC patients [64]. This contradiction is explainable. It is currently believed that the regulatory effect of VC on redox homeostasis is bidirectional; at physiological concentrations (plasma concentration: μM), it mainly exerts an antioxidant effect [65], while it tends to exert a pro-oxidative stress effect when the concentration is greatly increased (plasma concentration: mM) [66]. Therefore, increasing the intake of VC can raise the physiological concentration of VC and thereby exert antioxidant-related anticancer effects, whereas the application of high concentrations of VC aggravates cancer cell death caused by oxidative stress. Notably, the sensitivity of different cancers to high doses of VC is inconsistent [64,67], and fortunately, the significantly upregulated glycolysis in ccRCC drives its high sensitivity to VC-induced cellular oxidative stress damage [68]. Therefore, high-dose VC may hold promise as an adjuvant therapy for ccRCC patients. Similar to VC, UA, as an end product of purine catabolism [69], generally has antioxidant effects [70], but the relevance of UA metabolic homeostasis to cancer is controversial. For example, either too high or too low levels of UA can increase the risk of developing cancer [71]. Nonetheless, studies have found that UA overload is an important contributor to neoplastic death [72], and serum UA is associated with poor prognosis in ccRCC patients [73] as well as new-onset chronic kidney disease after nephrectomy [74]. Our study is the first to find elevated UA levels in ccRCC tissues, which may be a collateral effect of tumors preferentially utilizing fructose for energy supply under hypoxic conditions [75]. We speculate that the pro-inflammatory effects of UA may mediate the pro-oncogenic effects it exerts in ccRCC [76,77].

Third, our study found that RNS-related molecules (Arg, BH4, Cit, and GSNO) were upregulated in ccRCC tissues, while the correlation of ccRCC with RNS-related molecules was limited. Previous studies have shown that Arg deprivation therapy can effectively inhibit ccRCC proliferation and angiogenesis [78], suggesting that Arg may promote ccRCC progression. The role of BH4 in different tumors is contradictory [79]. Both the citrullinated histone process [80], which involves Cit, and the protein SNO modification process [81], which involves GSNO, were found to be associated with the prognosis of tumors. In addition, our study found that most of the lipid peroxidation products (4-HNE, 4-HHE, BEN, hexanal, MDA, 2,4.-Non, and 2,4.-Dec) were upregulated in the ccRCC group. This demonstrates that ccRCC undergoes lipid peroxidation [82] and suggests that manipulating lipid peroxidative damage may be a viable therapeutic approach for ccRCC.

Collectively, redox homeostasis reprogramming plays a complex and important role in the pathogenesis of ccRCC. We hypothesize that to combat the oxidative stress damage caused by rapid proliferation (e.g., RNS-related molecules and lipid peroxidation products), ccRCC cells significantly enhance their redox buffering capacity (e.g., redox couples) and antioxidant material reserves (e.g., endogenous redox molecules), thereby maintaining their redox homeostasis at a brand-new, superior level. Therefore, targeting redox homeostasis is a more promising therapeutic strategy for ccRCC than either antioxidative or pro-oxidant stress treatment alone.

Tumorigenic alterations in ccRCC manifest in three ways. First, there is an increase in ccRCC tissue heterogeneity. Both the cluster analysis and PCA results from proteomics and REME assays indicated that the heterogeneity within the ccRCC group was significantly higher than that of the paracancerous control group. Second, the GSVA results suggested that proximal tubular bicarbonate reclamation was significantly downregulated in the ccRCC group. Because ccRCC originates from the proximal tubular epithelium, the above results suggest a loss of the tissue properties of ccRCC. Finally, the pathways upregulated in the ccRCC group in GSVA and GSEA may be involved in shaping the malignant features of ccRCC. These include the cell cycle, focal adhesion pathway, and inflammation pathway (Toll-like receptor, chemokine, and Wnt signaling pathways), the former two being associated with the proliferative and metastatic properties of tumors, while the latter is associated with tumor progression and drug resistance [83]. Furthermore, ECM–receptor interactions, a core module enriched in differentially expressed SNO proteins, are also a cornerstone of tumorigenesis. Moreover, the organelles responsible for the process of protein translation and synthesis (spliceosomes and ribosomes), as well as the protein degradation process pathway (ubiquitin-mediated proteolysis), which are all upregulated in the ccRCC group, may be involved in supporting the metabolic reprogramming and redox homeostasis reprogramming of ccRCC.

The present study not only validates previous findings but also uncovers an entirely novel feature of ccRCC. However, there are also, inevitably, some limitations. First, our study had a small sample size, and all the patients had early-stage ccRCC, meaning that the characteristics of all the ccRCC patients cannot be represented. Second, the molecules targeted by REME technology are not comprehensive enough. Meanwhile, further mechanistic exploration of specific molecules is also lacking.

## 5. Conclusions

Our innovative multi-omics results provide an atlas of SNO proteins in ccRCC, identify some promising SNO proteins, and reveal that early ccRCC develops three major features typified by metabolic reprogramming, redox homeostasis reprogramming, and tumorigenic alterations. Oxidative stress damage caused by rapid proliferation together with an enhanced redox buffering capacity and antioxidant material reserves remodels the redox homeostasis of ccRCC. Redox species represented by NADPH/NADP^+^ are both good biomarkers and promising targets for targeted therapy.

## Figures and Tables

**Figure 1 antioxidants-12-00081-f001:**
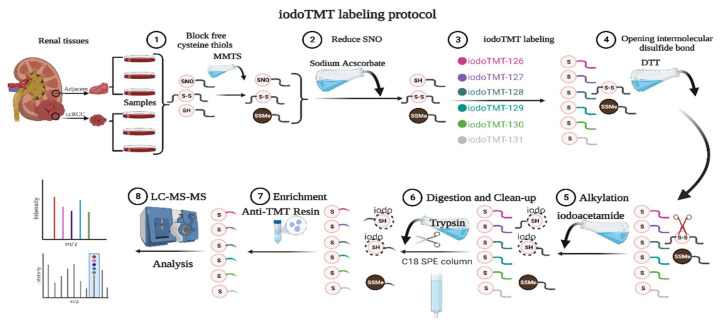
SNO-proteome protocol. MMTS: methylmethanethiosulfonate; SNO: protein S-nitrosylation; iodoTMT: iodoacetyl tandem mass tag; DTT: Dithiothreitol; LC-MS-MS: liquid chromatography-tandem mass spectrometry.

**Figure 2 antioxidants-12-00081-f002:**
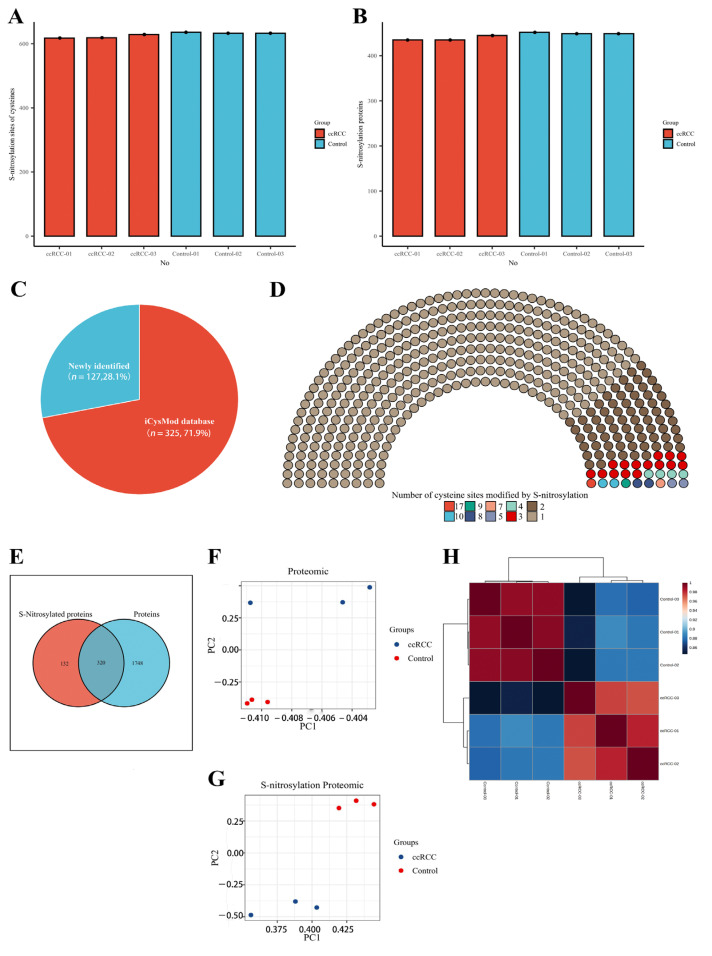
Proteomic and SNO−proteome landscape of ccRCC. (**A**) The number of SNO−cysteines detected in the ccRCC group and control group; (**B**) the number of SNO proteins detected in the ccRCC group and control group; (**C**) the proportion of newly identified SNO proteins and SNO proteins that have been confirmed by the iCysMod database; (**D**) number distribution of SNO−cysteine sites; (**E**) Venn diagram of proteins identified by the proteome and SNO−proteome; (**F**) PCA plot of the proteome; (**G**) PCA plot of the SNO−proteome; (**H**) sample correlation heatmap for the proteome.

**Figure 3 antioxidants-12-00081-f003:**
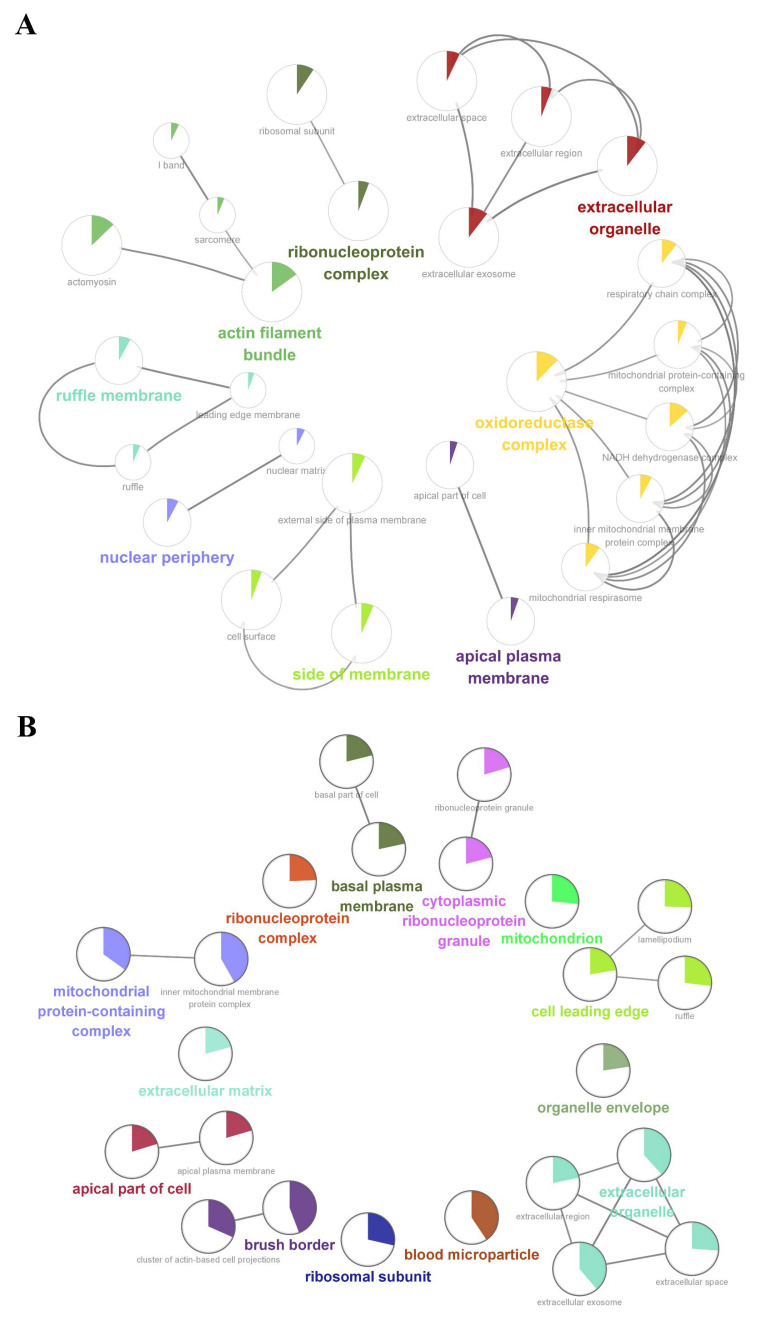
Cellular localization enrichment analysis of the SNO-proteome and proteome. (**A**) Cellular localization enrichment analysis of the SNO-proteome; (**B**) cellular localization enrichment analysis of the proteome.

**Figure 4 antioxidants-12-00081-f004:**
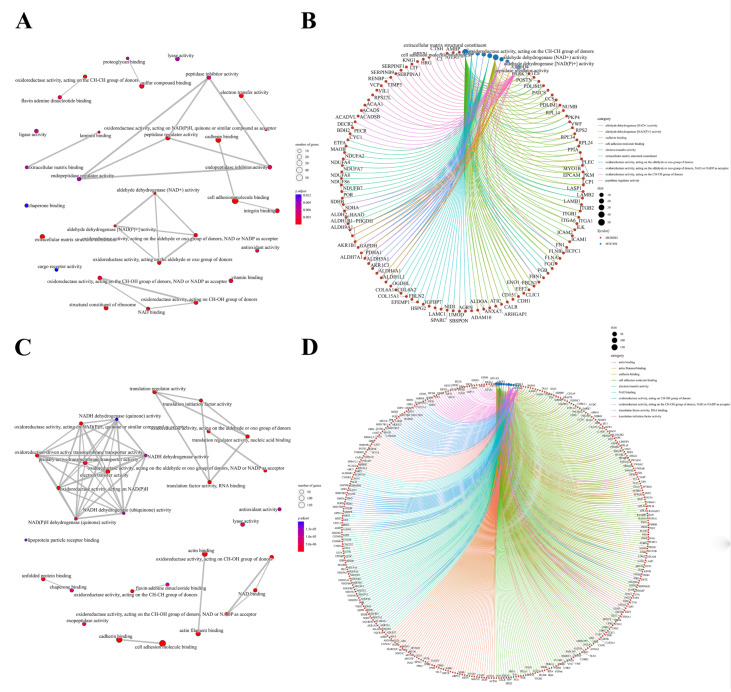
Molecular functions of the SNO-proteome and proteome. (**A**) Molecular function analysis result of the SNO-proteome; (**B**) molecular functional enrichment and protein clustering results of the SNO-proteome; (**C**) molecular function analysis result of the proteome; (**D**) molecular functional enrichment and protein clustering results of the proteome.

**Figure 5 antioxidants-12-00081-f005:**
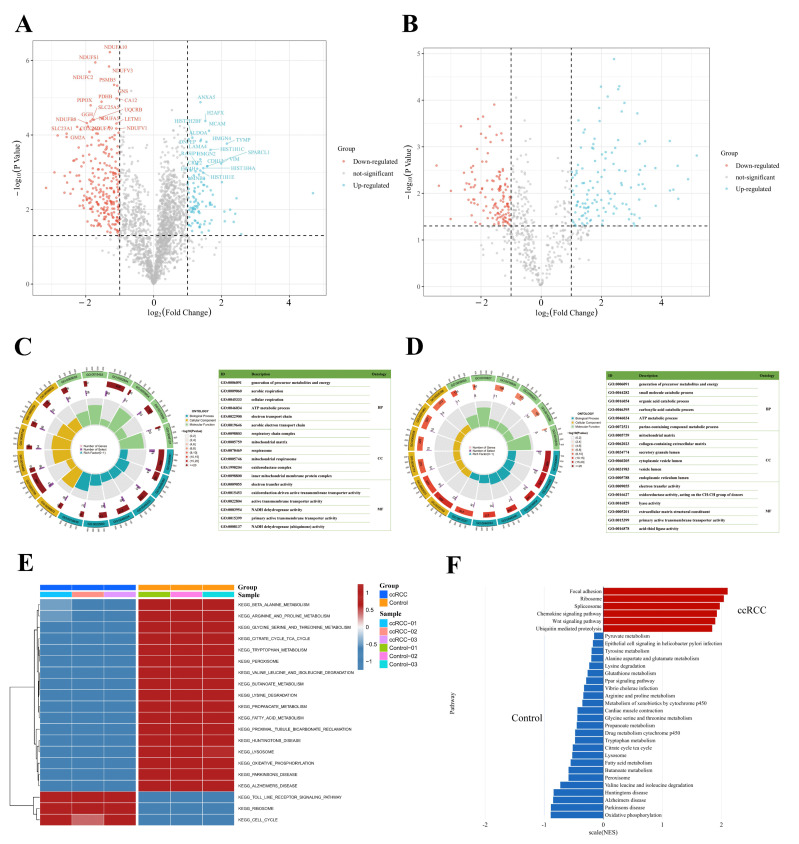
Integrated enrichment analysis of differentially expressed proteins and SNO proteins. (**A**) Heatmap of differentially expressed proteins; (**B**) heatmap of differentially expressed SNO peptides; (**C**) GO analysis of differentially expressed proteins; (**D**) GO analysis of differentially expressed SNO proteins; (**E**) GSVA analysis of differentially expressed proteins; (**F**) GSEA analysis of differentially expressed proteins.

**Figure 6 antioxidants-12-00081-f006:**
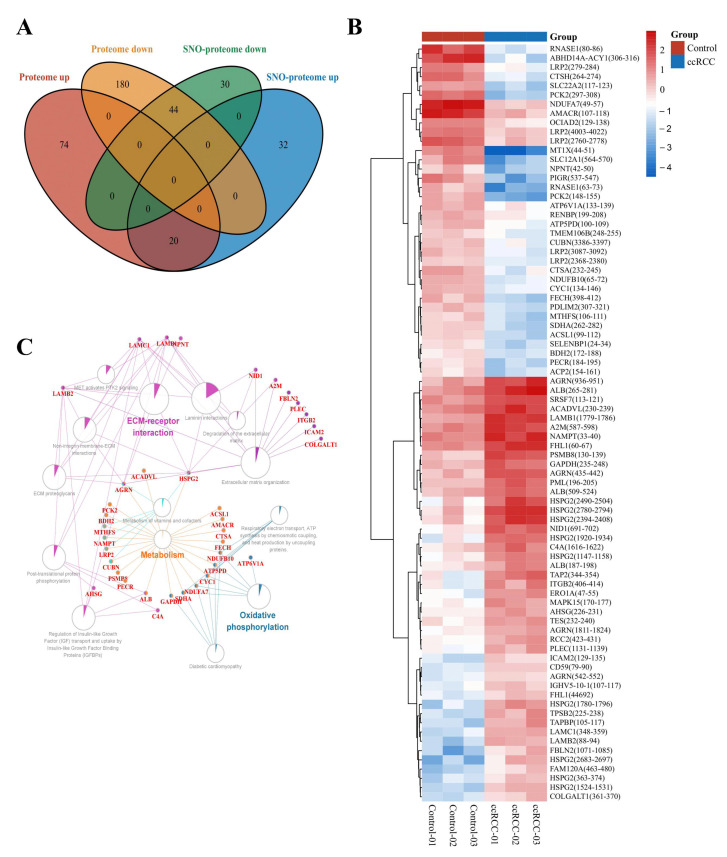
Molecular patterns of independently differentially expressed SNO proteins. (**A**) Venn diagram of differentially expressed SNO proteins vs. differentially expressed proteins; (**B**) heatmap of independently differentially expressed SNO−cysteine sites; (**C**) pathway−protein interaction network analysis of independently differentially expressed SNO proteins.

**Figure 7 antioxidants-12-00081-f007:**
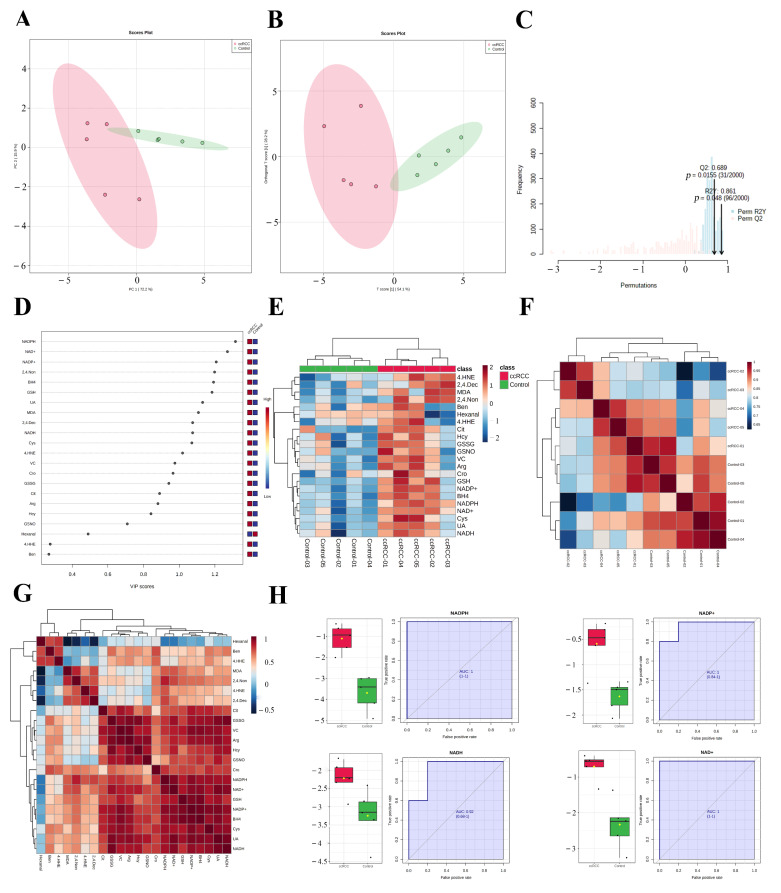
REME profiled the redox homeostasis reprogramming of ccRCC. (**A**) PCA of redox metabolites; (**B**) OPLS−DA of redox metabolites; (**C**) sevenfold cross−validation of the OPLS-DA model; (**D**) VIP scores of redox metabolites; (**E**) expression heatmap of redox metabolites; (**F**) Pearson correlation between samples; (**G**) Pearson correlation between redox metabolites; (**H**) ROC curves of NADH/NAD^+^ and NADPH/NADP^+^.

## Data Availability

The data are contained within this article and Appendix A.

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
