# Peer review of "Multi-Omics Approach Reveals Redox Homeostasis Reprogramming in Early-Stage Clear Cell Renal Cell Carcinoma"

_antioxidants, 2022, doi:10.3390/antiox12010081_

Round 1

Reviewer 1 Report

The manuscript “Multi-Omics Approach Reveals Redox Homeostasis Reprogramming in Early-Stage Clear Cell Renal Cell Carcinoma”, by Zhang and collaborators deals with an important issue of molecular mechanisms underlying the clear cell renal carcinoma, in particular redox homeostasis characteristics and protein S-nitrosylation modification. Although authors used relatively small sample (five samples in control and tumor group), concerning exhaustive and detail characterization of redox homeostasis parameters, thei obtained consistent results, which contribute substantially to our knowledge on ccRCC biology. The paper is well organized and clearly written and is easy to read it. It gives systematic presentation of and reasonable explanation of the results obtained. A manuscript perfectly fits into the scope of the journal. Hence, the manuscript could be interesting for readers and I recommend it for publication after minor revision.

Minor issues for improvement:

- Some links referring methodology are not valuable anymore, they should be refreshed.

- Graphs are very difficult if not impossible to read in order to make conclusions, resolution and font should be changed. If not (because of the limitations of software and final illustration size), could original illustrations been added as supplementary files?

- Material and methods (669-70): “Limited by the volume of kidney specimens, omics testing of each sample was completed with only one biological replicate”. – If I understood well, they had 5 biological replicates (and results for 5 specimens are presented in most of the illustrations). Did authors mean “technical replicates”?

Author Response

- Some links referring methodology are not valuable anymore, they should be refreshed.

Re: Thanks for your suggestions, we update the references below. 【17】【20】【24】【25】

- Graphs are very difficult if not impossible to read in order to make conclusions, resolution and font should be changed. If not (because of the limitations of software and final illustration size), could original illustrations been added as supplementary files?

Re: Thank you for your suggestions. We have provided high-resolution images in the supplementary files.

- Material and methods (669-70): “Limited by the volume of kidney specimens, omics testing of each sample was completed with only one biological replicate”. – If I understood well, they had 5 biological replicates (and results for 5 specimens are presented in most of the illustrations). Did authors mean “technical replicates”?

Re: Thank you for your correction. It should indeed be revised as "technical replicate". We have updated it in the manuscript.

Thank you for your support of our research.

Merry Christmas~

Reviewer 2 Report

This work is original and very dense in terms of analysis. It is very easy to read, which shows the quality of the writing. In my field of expertise, I do not ask for any correction, so I give a favourable opinion for a publication as it is.

The work of Zhang et al. is essential because it is based on an in situ study of tumour samples versus normal samples close to the tumour site of 5 patients. Furthermore, the localisation and functional properties of SNO proteins in ccRCC tumours and paracancerous normal samples were elucidated for the first time. The results presented by Zhang et al. and illustrated by high quality figures will help the scientific community to better understand the pathogenesis of ccRCC and surely in the future to identify new patients to be seriously considered for the diagnosis and accurate therapy of ccRCC.

Author Response

Re: Thank you for your great support and affirmation of our study!

Merry Christmas!

Reviewer 3 Report

The tumor and adjacent tissues obtained were frozen until use. Authors are encouraged to include information regarding sample handling from the point of origin to processing and freezing. This is an important consideration since there may be some artifacts that are introduced as a result of sample handling from a hypoxic environment to the atmospheric oxygen conditions. What was done to maintain the fresh tissue oxygen levels? 

Author Response

The tumor and adjacent tissues obtained were frozen until use. Authors are encouraged to include information regarding sample handling from the point of origin to processing and freezing. This is an important consideration since there may be some artifacts that are introduced as a result of sample handling from a hypoxic environment to the atmospheric oxygen conditions. What was done to maintain the fresh tissue oxygen levels?

Re: Thank you for your suggestions.

First, we added the original processing details of the samples as well as relevant citations according to your suggestions (lines: 66-67).

At the same time, we note that the question you asked is very important, especially when looking at redox-related issues.

Through a literature search, we found that most of the current research was mainly conducted by liquid nitrogen snap freezing and dry ice manipulation to minimize the influence of the external environment on the samples.

In our study,

1: All tissue specimens were obtained by laparoscopic surgery: laparoscopic surgery would use carbon dioxide to maintain intra-abdominal pressure, which was also beneficial to maintain the low oxygen status of the samples when they were excised within the abdominal cavity.

2: After kidney tissue ex vivo, we completed the procedure for liquid nitrogen freezing in a very short time (within 1min), which should minimize the impact of the atmosphere on the sample.

3: Tumor and control tissues were treated the same way and for the same time and the impact of the atmosphere affected should be similar.

Finally, we are very eager that you can make valuable suggestions to guide us to better maintain the oxygen level of fresh tissue in the future, which will make our research better.

Thank you again for your guidance!

Merry Christmas~

Round 2

Reviewer 3 Report

I am satisfied with the author's clarification to the question raised.